# Entangled Connections: HIV and HPV Interplay in Cervical Cancer—A Comprehensive Review

**DOI:** 10.3390/ijms251910358

**Published:** 2024-09-26

**Authors:** Giuliana Pavone, Andrea Marino, Viviana Fisicaro, Lucia Motta, Alessandra Spata, Federica Martorana, Serena Spampinato, Benedetto Maurizio Celesia, Bruno Cacopardo, Paolo Vigneri, Giuseppe Nunnari

**Affiliations:** 1Department of Clinical and Experimental Medicine, University of Catania, 95123 Catania, Italy; giuliana.pavone@humanitascatania.it (G.P.); luciamotta693@gmail.com (L.M.); federica.martorana@unict.it (F.M.); vigneripaolo@gmail.com (P.V.); 2Medical Oncology Unit, Humanitas Istituto Clinico Catanese, 95045 Catania, Italy; 3Unit of Infectious Diseases, Department of Clinical and Experimental Medicine, ARNAS Garibaldi Hospital, University of Catania, 95123 Catania, Italy; bmcelesia@gmail.com (B.M.C.); cacopard@unict.it (B.C.); giuseppe.nunnari1@unict.it (G.N.); 4Department of Clinical and Experimental Medicine, University of Messina, 98124 Messina, Italy; viviana.fisicaro@gmail.com (V.F.); serenaspampinato93@gmail.com (S.S.); 5Medical Oncology Unit, Department of Human Pathology “G. Barresi”, University of Messina, 98124 Messina, Italy; alessandraspata@gmail.com

**Keywords:** HIV, HPV, AIDS, cervical cancer, vaccines, antiretroviral therapy

## Abstract

Cervical cancer (CC) remains a prevalent malignancy and a significant global public health concern, primarily driven by persistent human papillomavirus (HPV) infections. The infectious nature of HPV underscores the preventability of CC through vaccination and screening programs. In addition to HPV, factors such as age, parity, smoking, hormonal contraceptives, and HIV co-infection elevate the risk of CC. HIV-associated immunodeficiency exacerbates susceptibility to infections and cancers, making CC a defining condition for acquired immune deficiency syndrome (AIDS) and one of the most commonly diagnosed cancers among women living with HIV (WLWH). These women face higher risks of HPV exposure due to sexual behavior and often encounter economic, social, and psychological barriers to screening. HIV and HPV co-infection can potentially accelerate CC carcinogenesis, with WLWH typically being diagnosed with CC earlier than their HIV-negative counterparts. Antiretroviral therapy (ART), which reduces AIDS-related mortality, also lowers the risk of invasive CC. The interaction between HIV and HPV is intricate and bidirectional. This summary reviews current evidence on HPV infection and CC in WLWH, highlighting the connections across pathogenesis, prevention, diagnosis, and treatment.

## 1. Introduction

Cervical cancer (CC) is a prevalent malignancy among women and a significant global public health issue, primarily caused by persistent infection with human papillomavirus (HPV) [1]. Due to the infectious trigger, CC is largely preventable through primary prevention measures, such as vaccination, and secondary prevention methods, such as regular screenings [2]. Besides HPV, several other factors could enhance the risk of developing CC, including age, parity, smoking status, hormonal contraceptive use, and co-infection with human immunodeficiency virus (HIV) [1]. Indeed, HIV-related immunodeficiency increases the risk of cancers, both AIDS and non-AIDS related [3,4,5]. Consequently, CC is classified as an acquired immune deficiency syndrome (AIDS)-defining condition and is the most frequently diagnosed cancer among women living with HIV (WLWH) [6]. WLWH face a higher risk of HPV exposure due to certain sexual behaviors and often exhibit lower adherence to screening programs, driven by economic, social, and psychological barriers [7,8]. Furthermore, the co-infection of HIV and HPV facilitates cervical carcinogenesis through a complex viral interplay. As a result, WLWH are diagnosed with CC approximately 5 to 10 years earlier than women without HIV [2,9]. Moreover, antiretroviral therapy (ART), which has significantly decreased AIDS-related mortality, is also linked to a reduced risk of invasive CC [10]. Overall, the interaction between HIV and HPV infection is complex, multifactorial, and bidirectional.

Here, we summarize the current evidence on HPV infection and CC in WLWH, focusing on prevention, diagnosis, treatment, and prognosis of this AIDS-defining disease.

## 2. Incidence and Prevalence of Cervical Cancer in WLWH

According to the World Health Organization (WHO) approximately 660,000 new cases of CC have been diagnosed in 2022, with around 350,000 resulting in death, making CC the fourth leading cause of cancer-related death in women [11].

The incidence of CC, like that of HIV infection, varies significantly across different regions of the world, reflecting disparities in healthcare resources and prevention strategies [12]. In high-income countries (HICs), the incidence and mortality rates of cervical cancer have significantly declined over the past five decades. This success is largely due to the widespread implementation of HPV vaccination programs, which have drastically reduced the prevalence of high-risk HPV types associated with CC. Additionally, regular cervical screening programs, such as Pap smears and HPV DNA testing, have been instrumental in the early detection and treatment of precancerous lesions, further reducing the risk of progression to invasive cancer [2,13]. As a result, HICs have seen a remarkable decline in CC incidence and mortality, showcasing the effectiveness of a well-coordinated public health response [2,13]. Conversely, low- and middle-income countries (LMICs) face substantial challenges in combating CC. In these regions, particularly in sub-Saharan Africa, limited healthcare resources, low public awareness, and inadequate screening and vaccination coverage contribute to the high burden of CC [14]. The situation is exacerbated by the high prevalence of HIV in these regions. WLWH are particularly vulnerable, with the incidence of CC among this population estimated to be around 85%, reflecting a significantly elevated risk [2]. In fact, WLWH are approximately six times more likely to develop cervical cancer compared to their HIV-negative counterparts, making HIV a critical risk factor for the disease [12,15,16]. While the introduction of antiretroviral therapy (ART) has contributed to a reduction in CC incidence among WLWH, particularly in HICs, the decrease has not been as substantial as that seen with other AIDS-defining cancers, such as Kaposi’s sarcoma and certain lymphomas. This suggests that while ART has improved the overall health and longevity of WLWH, it has not fully mitigated the elevated risk of CC in this population. Consequently, there remains an urgent need for enhanced prevention, early detection, and treatment strategies in regions with high HIV prevalence, especially within LMICs [2,17,18,19]. Addressing these disparities is crucial to reducing the global burden of CC and achieving more equitable health outcomes for women worldwide.

## 3. Molecular Mechanisms of Oncogenesis in HPV-Related Cervical Cancer

The interplay between human papillomavirus (HPV) oncogenic proteins E5, E6, and E7 and HIV profoundly influences the pathogenesis of CC in co-infected individuals (Table 1). These proteins are central to HPV’s ability to induce malignant transformation and are particularly impactful under the compromised immune conditions caused by HIV [20,21,22].

E5 Protein: The E5 protein of HPV plays a crucial role in augmenting the growth and survival signals within the host cell. By enhancing signaling pathways such as the epidermal growth factor receptor (EGFR) pathway, E5 promotes cellular proliferation and helps in the maintenance of the malignant state. In HIV-infected patients, where cellular immunity is compromised, the role of E5 in promoting oncogenesis becomes more pronounced, as the immune system is less capable of containing the unregulated growth induced by this protein. The interaction of E5 with host immunity suggests a potentiated oncogenic environment in the context of HIV co-infection [23].

E6 Protein: E6 is perhaps the most notorious of the HPV oncogenic proteins due to its ability to bind to and promote the degradation of the tumor suppressor protein p53. This interaction leads to the inhibition of apoptosis, allowing HPV-infected cells to survive and accumulate genetic damage. In the setting of HIV co-infection, the efficacy of the immune system to clear such damaged cells is diminished, which may lead to an increased risk of malignant transformation and cancer progression. Furthermore, studies have shown that HIV proteins like Vpr can modulate the activity of molecules involved in the p53 pathways, thereby enhancing the oncogenic potential of E6 [24,25].

E7 Protein: The E7 protein contributes to oncogenesis by binding to the retinoblastoma (pRb) family of proteins, disrupting their function and thereby driving the cell cycle progression inappropriately. This disruption is crucial for viral replication and persistence but also leads to increased cellular turnover and potential malignant transformation. In HIV-positive individuals, the regulatory oversight over cell cycle progression may be further compromised due to systemic immune suppression, enhancing the impact of E7 on cervical epithelial cell transformation [24,25,26,27,28,29,30].

## 4. Interplay between HIV and HPV in Cervical Cancer Development

### 4.1. Immune System Dysfunction in HIV and HPV Co-Infection

HPV and HIV, although distinct in their characteristics and modes of infection, exhibit interconnected pathogenetic mechanisms that influence the progression of both infections and related diseases. The pathogenesis of HPV and HIV infections and their interaction represent a complex and crucial theme for understanding the increased risk of CC in coinfected women. Several studies have explored how these infections influence the immune system and the progression of precancerous and cancerous lesions. Mondatore et al. [31] highlighted that WLWH have a significantly higher risk of developing high-grade squamous intraepithelial lesions (HSILs), especially when infected with hrHPV types. The persistence of HPV, facilitated by HIV-induced immunosuppression, has been identified as the primary risk factor for developing HSIL [32]. The dysfunction of T lymphocytes specific for HPV oncoproteins, induced by HIV, has been identified as a key factor contributing to the increased risk of CC. Women living with HIV show reduced frequency and magnitude of T-cell responses specific to HPV E6 and E7 oncoproteins. This immune dysfunction is more pronounced in patients with advanced HIV disease, suggesting that prolonged and inadequately treated immunodeficiency may exacerbate cervical lesion progression [24,33,34]. HIV targets CD4+ T cells, leading to apoptosis and cell depletion. Markers of CD4+ T-cell apoptosis, including TRAIL, CCR5 microparticles, TNFR2, and soluble FAS ligand, increase with increasing HIV viremia. CD4+ T-cell depletion leads to an immunocompromised state and increases susceptibility to disease progression caused by other coinfections, oncogenesis, and associated viral neuropathogenesis. Dendritic cells (DCs) play a key role in antigen presentation and T-cell activation. In the cervical tissues of individuals coinfected with HIV and HPV, a significant increase in DCs expressing the CD1a marker has been found, although many of these cells showed an immature phenotype characterized by low CD83 and CD86 expression. This could limit their ability to activate an effective immune response against HPV, contributing to persistent infection [35]. DCs are therefore an emerging research target, as they are involved in the initial detection and response to primary viral infection [36]. They not only are able to detect HIV-1 through both CD4 and dendritic cell-specific ICAM-3 grabbing non-integrin (DC-SIGN) but can also be infected by HIV-1, transmitting the virus to CD4+ T cells and exploiting the immune response to its advantage. DC-SIGN on the surface of DCs acts as a PRR for mannose patterns on the gp120 site of HIV-1 [37]. Whether an adaptive response is initiated depends on the structure of N-glycans in the HIV-1 pathogen. The greater the mannose composition of HIV-1 N-glycans, the more effective the binding of HIV-1 to DC-SIGN for adaptive immune purposes [38]. The DC cell is transported to various parts of the body via chemokines and their receptors. These APCs move towards lymph nodes, including those of the central nervous system, as well as gut-associated lymphoid tissues. In the lymph nodes, exposed DCs mature and present antigenic peptides on MHC molecules, activating the adaptive immune response. During viral infections and inflammatory conditions, activated T cells and antigen-presenting cells express high levels of class II major histocompatibility complex (MHC) molecules (HLA-DR).

In the cervical tissues of individuals co-infected with HIV and HPV, an elevated expression of HLA-DR compared to controls has been observed. This upregulation of MHC expression may affect the presentation of viral antigens and the activation of local immune responses. Both B and T cells are activated, with B cells producing antibodies and T cells targeting infected cells. Unfortunately, the adaptive immune response is insufficient to clear the infection, often being too delayed and inadequate [39,40]. Mature dendritic cells (DCs) play a crucial role in mounting an immune response against HPV. In the tumor environment influenced by high-risk HPV, DCs play a pivotal role but exhibit compromised functionality due to HIV co-infection [41]. Specifically, DCs in co-infected individuals may attenuate the functions of effector T cells through immunoinhibitory signaling involving PD-1/PD-L1 pathways [42]. This interaction leads to a loss of cytotoxic T lymphocyte (CTL) function, a critical component in fighting cancer cells. HIV exacerbates this effect by further diminishing the immune response capability, which is already compromised by HPV. Moreover, cervical tumor cells may inhibit DC migration to lymph nodes through CCR7 depletion and induce DCs to produce matrix metalloproteinase 9 (MMP-9), further promoting tumor establishment. This scenario is particularly detrimental in HIV-infected patients where immune dysfunction allows for enhanced viral oncogenesis facilitated by HPV oncoproteins like E6 and E7, driving the progression toward malignancy [35,43,44].

Furthermore, DCs are important mediators of innate and adaptive immunity. They are among the first cells to encounter viral infection and thus play an integral role in responding to and propagating infection in various viral infections [45]. Analyzing the relationship between HIV/AIDS and the persistence of hrHPV, which can lead to CC, it is crucial to evaluate the role of plasmacytoid dendritic cells (pDCs) and regulatory T cells (Tregs) in the persistence of hrHPV n HIV-infected and non-infected women [46]. Plasmacytoid dendritic cells, involved in both innate and adaptive immunity, show a decrease in HIV-positive patients, correlated with opportunistic infections. Tregs, which negatively regulate immune responses, are present at elevated levels in HIV-positive individuals. Therefore, low pDC levels and high Treg levels can be significantly associated with the persistence of hrHPV in both HIV-positive and -negative women, suggesting the need for broader studies, as this increase in Tregs is associated with immunosuppression that favors the progression of CC [47].

The vaginal microbiome and local inflammatory response are also emerging as key factors in shaping the immune response to viral infections. A healthy vaginal microbiota not only maintains an acidic environment that inhibits viral transmission but also plays a critical role in modulating immune activity within the genital tract [48]. Increasing evidence suggests that the vaginal microbiome directly impacts the co-infection dynamics of HIV and HPV by regulating immune activation and controlling inflammation, both of which are central to viral persistence and replication [49]. Dysbiosis, or disruption of the microbiota balance, often leads to heightened inflammation and immune dysregulation, creating a favorable environment for viral infections. Although the precise mechanisms by which the microbiome affects immune homeostasis in the context of HIV, HPV, and other sexually transmitted infections remain under investigation, its role in mitigating infection progression is becoming increasingly clear [50]. Further studies are needed to better understand how the vaginal microbiome shapes the immune landscape in these contexts.

### 4.2. Epithelial–Mesenchymal Transition in HIV and HPV-Associated Cervical Cancer

HIV infection, through the gp120 and Tat proteins, can induce epithelial–mesenchymal transition (EMT) in cervical tissue, promoting the progression of cervical lesions. EMT is a physiological process that regulates differentiation and the acquisition of cellular lineage identity during embryonic development. However, in cancer, EMT facilitates tumor progression by promoting the growth and metastasis of invasive epithelial tumor cells [51].

Cancer-associated EMT is a multi-step process that starts with the loss of apicobasal polarity in epithelial cells, followed by the breakdown of adherens and tight junctions, a decrease in epithelial markers, and a subsequent loss of cellular adhesion [52]. The role of transforming growth factor beta (TGF-β) as a key regulator of EMT in cancer is well established [53]. Recent studies suggest that inhibiting the MAPK and TGF-β signaling pathways may limit HIV-1-induced EMT [54].

HIV-1 gp120 and Tat proteins have also been shown to increase the invasiveness of both HPV-infected and non-infected epithelial tumor cells by enhancing their migratory and invasive potential, thus contributing to metastasis and drug resistance [55]. In HIV-positive patients, a significant reduction in epithelial markers such as E-cadherin and cytokeratin, alongside an increase in mesenchymal markers like N-cadherin and vimentin, suggests a heightened level of EMT. Transcriptome sequencing of treated cells revealed extensive gene expression alterations, with the activation of the Wnt/β-catenin signaling pathway, which is crucial for EMT regulation [56,57,58].

The HPV E5 protein also plays a pivotal role in promoting EMT by driving a switch in FGFR2 isoforms, leading to the downregulation of the epithelial FGFR2b isoform and the aberrant expression of the mesenchymal FGFR2c variant. This shift fosters tumorigenic behavior and impairs normal cellular differentiation [59,60].

In summary, HIV infection can accelerate cervical lesion progression by inducing EMT through the activation of the Wnt/β-catenin pathway by gp120 and Tat proteins. Modulating regulatory T cells (Tregs) and plasmacytoid dendritic cells (pDCs) could provide an effective strategy for managing persistent oncogenic HPV infections and related neoplasia, particularly in high-risk HIV-positive patients.

Recent research has illuminated the complex molecular interactions between HIV and HPV that exacerbate the oncogenic potential of HPV in the presence of HIV co-infection (Table 2). The HIV Tat protein enhances the transcriptional activity of HPV oncogenes E6 and E7 by increasing their promoter activities, thereby upregulating their expression, which is crucial for oncogenic transformation of host cells [54]. Additionally, the HIV Vpr protein disrupts the cellular G2/M checkpoint control, facilitating the accumulation of mutations in HPV-infected cells and promoting their progression to malignancy [61]. These direct interactions are compounded by HIV-induced immune suppression, notably the impairment of CD4+ T cells and DCs, which diminishes the host’s ability to mount effective immune responses against HPV [35]. This immune suppression leads to decreased surveillance and clearance of HPV-infected cells, allowing for persistent infections that are more likely to progress to cancer [31]. Epidemiologically, the co-infection with HIV is associated with a higher prevalence of high-risk HPV types and a greater incidence of cervical precancerous lesions, suggesting a synergistic interaction at the molecular level that enhances HPV’s pathogenicity [12].

As discussed by Lien et al. [42], HIV exacerbates the risk of developing HPV-associated tumors by weakening immune responses and directly impacting epithelial cells. Contact between oral or genital mucosal cells and HIV-1 virions, as well as gp120 and Tat proteins, leads to cellular junction dysfunction and EMT induction, accelerating the progression toward malignancy.

During the final stages of EMT, cells acquire a spindle cell morphology and express mesenchymal markers. Cells in intermediate stages of EMT can express both epithelial (E-cadherin) and mesenchymal (vimentin) markers, presenting a hybrid phenotype, a critical factor contributing to tumor cell invasiveness. E-cadherin and CD44 are glycoproteins involved in cellular adhesion. Their alterations are associated with tumor progression and invasiveness [62]. In examining the differences in E-cadherin and CD44 adhesion protein expression in cervical intraepithelial neoplasia (CIN) between HIV-positive and HIV-negative women, and evaluating the correlation with HPV infection, various studies have found that alterations in E-cadherin and CD44 expression are common in CIN and are influenced by HPV infection [63]. HIV-positive women have a higher risk of HPV infection and CIN, characterized by rapid progression and frequent recurrences [63]. Alterations in E-cadherin and CD44 expression are closely correlated with the severity of CIN [63]. The HIV Tat protein induces EMT in HPV-infected epithelial cells, increasing their invasiveness. This process is mediated by activating the TGF-β and MAPK signaling pathways, leading to reduced E-cadherin expression and increased vimentin and N-cadherin expression. Inhibiting these pathways can limit HIV-induced EMT development [54,55,63,64].

The local immune response is crucial for the regression of HPV lesions: HIV infection alters the local cytokine profile, worsening the natural course of HPV infection (Table 2). Langerhans cells, crucial for the local immune response, are reduced in CIN lesions in HIV-positive patients, further compromising local immunity [65]. The production of anti-HPV antibodies is increased in HIV-positive patients but does not seem sufficient to eliminate the HPV infection. Various studies examined the influence of HIV–HPV coinfection on the distribution of immune cells and cytokine production in the cervix. It was observed that coinfected women showed a significant reduction in cells expressing pro-inflammatory cytokines like IL-6, TNF-α, and IFN-γ compared to women infected only with HPV. This reduction could explain the increased persistence of HPV infections in WLWH, contributing to the progression to precancerous lesions and CC [35,66]. In women coinfected with HIV and HPV, increased production of pro-inflammatory cytokines such as IL-1α and IL-1β has been observed. However, these responses do not always result in reduced viral replication, suggesting that chronic inflammation might facilitate viral persistence and disease progression [35,46] (Figure 1).

The outer ring highlights HIV and HPV infections as key initial risk factors. HPV is the primary cause of cervical cancer, while HIV increases the risk by compromising the immune system. The middle ring illustrates how immune dysfunction and the epithelial mesenchymal transition (EMT) lead to the progression of HPV infection. HIV weakens the immune response, facilitating HPV persistence, which is crucial for the development of precancerous lesions. The inner ring details the progression from precancerous lesions to invasive tumors. Persistent HPV infection can lead to cervical intraepithelial neoplasia (CIN) and, eventually, invasive cervical cancer. At the center lies invasive cervical cancer, the final outcome of the interplay between risk factors, immune impairment, and lesion progression.

## 5. Screening and Management of Precancerous Cervical Lesions in WLWH

According to the latest WHO guidelines, HPV DNA testing every 5 to 10 years is the primary method for CC screening for the general population in high-income countries (HICs), due to its high sensitivity when used with the Pap test [11,67,68,69]. WLWH deserve a more intense screening program, due to the higher risk of developing precancerous cervical lesions and CC. For WLWH aged 25–49, HPV DNA testing is recommended every 3 to 5 years [11]. Accordingly, recent guidelines from the American Society of Clinical Oncology (ASCO) recommend screening WLWH with HPV DNA testing twice as often as HIV-negative women [70]. Overall, HPV DNA testing exhibits comparable sensitivity but lower specificity in the detection of high-grade cervical abnormalities in WLWH compared to HIV-negative women [10,71]. The management of cervical precancerous lesions is also impacted by HIV infection, and several studies have shown poorer treatment outcomes in WLWH compared to HIV-negative women, regardless of the method employed [72]. In HICs, excisional procedures, such as loop electrosurgical excision, large loop excision of the transformation zone, or cone excision, are the primary treatment modalities for histologically confirmed grade 2 and 3 cervical intraepithelial neoplasia (CIN2/3) or adenocarcinoma in situ (AIS). WLWH usually present with larger precancerous lesions, leading to a higher likelihood of positive margins and recurrence compared to HIV-negative women [72,73,74,75,76]. Additionally, WLWH are more likely to present cervical abnormalities ineligible for ablation, which must be treated with excision [2,77]. Although excisional treatments may be more effective than ablative treatments in WLWH, they are more invasive and display a higher risk of procedural complications and sequalae, including preterm birth [77,78,79].

## 6. Benefits of HPV Vaccination in WLWH

Primary prevention through HPV vaccination is crucial for WLWH due to their heightened vulnerability to persistent HPV infections and HPV-related CC. WLWH are at a higher risk for persistent HPV infections due to their compromised immune systems, particularly those with low CD4 counts, which can lead to a higher incidence of CIN and CC [80,81,82]. The immunogenicity of HPV vaccines in WLWH has been found to be robust, with studies indicating good seroconversion rates, although slightly lower than in HIV-negative individuals. This indicates that the vaccine can elicit a protective immune response, thereby reducing the prevalence of high-risk HPV (hrHPV) infections and subsequent cervical disease [80,83]. HPV vaccination has demonstrated substantial efficacy in preventing hrHPV infections, which is pivotal for reducing CC incidence among WLWH. Furthermore, the HPV vaccine has shown to significantly reduce the incidence of genital warts and other HPV-related cancers, such as anal and oropharyngeal cancers, which WLWH are also at increased risk for due to their immunocompromised status [84]. Effective vaccination can lead to a decrease in the overall burden of HPV-related diseases, improving quality of life and reducing healthcare costs associated with the management and treatment of these conditions [80,83]. Despite the proven benefits, barriers such as vaccine accessibility, awareness, and healthcare infrastructure pose challenges, especially in sub-Saharan Africa, where the burden of both HIV and CC is high. Implementing comprehensive HPV vaccination programs tailored to the needs of WLWH can significantly mitigate these risks and advance public health goals [81,82,85].

## 7. Role of ART in the Treatment of Precancerous Lesions and Invasive Cervical Cancer

In WLWH, a low CD4 count is strongly associated with the persistence of HPV infection and progression to precancerous lesions and eventually to CC [2,86]. Upon ART initiation, achieving rapid virological control and maintaining treatment adherence led to better mucosal immune reconstitution, improving long-term health outcomes for WLWH [10,87,88]. A meta-analysis of cross-sectional and cohort studies assessed the risk of HPV infection and cervical lesions in women using antiretroviral therapy (ART) compared to ART-naive women. Overall, ART was associated with a reduced incidence of high-grade cervical abnormalities (adjusted odds ratio [OR] 0.59), a slower progression from low-grade to high-grade cervical abnormalities (adjusted OR 0.64), and an increased rate of cervical abnormality regression (adjusted hazard ratio [HR] 1.54) [10]. Additionally, two studies included in the meta-analysis demonstrated that ART was associated with a reduced risk of CC (adjusted HR 0.50) [89,90]. However, the effect of ART was not homogeneous under a geographical standpoint: Cohorts from Africa, Europe, and North America confirmed the benefit of ART on cervical lesions, while studies from Latin America and Asia showed an increased risk of cervical abnormalities despite ART, likely due to older guidelines with lower CD4 thresholds for antiviral treatment initiation [10]. While the role of early ART initiation in promoting regression of precancerous cervical lesions is well established, its effectiveness in treating CC remains poorly understood. According to current guidelines, concurrent chemoradiation is the standard of care for invasive CC in patients with stage IB2-IVA disease [91,92]. In recent years, several trials have advanced the therapeutic algorithm for invasive CC. The GOG-240 trial demonstrated improved survival with the addition of the antiangiogenic agent bevacizumab to platinum compounds and paclitaxel in recurrent, persistent, or metastatic CC [93]. The KEYNOTE-826 trial highlighted the role of the immune checkpoint inhibitor pembrolizumab [94]. Based on this study, pembrolizumab, a monoclonal antibody against PD-L1, is recommended in addition to chemotherapy with or without bevacizumab as first-line therapy for persistent, recurrent, or metastatic CC with a PD-L1 combined positive score (CPS) of greater than 1 [92]. More recent randomized controlled trials, such as the INTERLACE and KEYNOTE-A18 studies, aim to evaluate the efficacy of chemoradiotherapy treatments enhanced with either induction chemotherapy or the addition of immunotherapy, respectively, to improve outcomes for patients with CC across both early and advanced stages [95,96]. In WLWH and CC, the implementation of chemoradiation-based treatments could be challenging, primarily due to concerns about immune status. HIV infection leads to the depletion of CD4 lymphocytes, resulting in suppressed immunity and an increased risk of opportunistic infections [97]. With the advent of ART, there has been considerable improvement in the immune status of WLWH [98]. A retrospective cohort study of 4314 women diagnosed with CC in South Carolina from 1998 to 2018, including 53 WLWH, found that survival times were similar between WLWH and HIV-negative women [99]. Different studies showed that chemo-radiation can be well tolerated by WLWH in treatment with ART, whereas others reported increased toxicities in this population, especially hematologic and gastrointestinal toxicities [100,101,102,103]. Notably, many antiretrovirals and anticancer agents are known to interact with each other and with other drugs and molecules, as they are often metabolized through the same enzymatic pathways [104,105,106]. In addition, these patients may exhibit poor treatment compliance due to factors such as inadequate nutritional status and insufficient social support [107].

## 8. Conclusions

CC remains a significant global health challenge, particularly among WLWH. The intricate relationship between HPV and HIV exacerbates the risk and progression of CC, necessitating targeted prevention and treatment strategies. Despite advances in HPV vaccination and ART, significant disparities persist, particularly in low- and middle-income countries, where limited access to healthcare resources and prevention programs poses ongoing challenges. High-risk HPV genotypes, such as HPV-16 and HPV-18, are key drivers of cervical carcinogenesis through the expression of oncoproteins E6 and E7, which interfere with crucial tumor suppressor pathways. The immunocompromised state of WLWH further complicates the clinical landscape, as it facilitates persistent HPV infections and accelerates the progression of precancerous lesions. While ART has been instrumental in reducing the incidence of CC in some regions, its impact on WLWH is varied, underscoring the need for more nuanced approaches to management. Primary prevention through HPV vaccination, tailored screening programs, and timely treatment of precancerous lesions are critical components in combating CC. Additionally, integrating novel therapies, such as immune checkpoint inhibitors and antiangiogenic agents, with standard treatment protocols holds promise for improving outcomes in invasive CC. To effectively address the burden of CC, especially among WLWH, global efforts must focus on improving healthcare infrastructure, enhancing access to preventive measures, and supporting ongoing research into innovative therapeutic interventions.

## Figures and Tables

**Figure 1 ijms-25-10358-f001:**
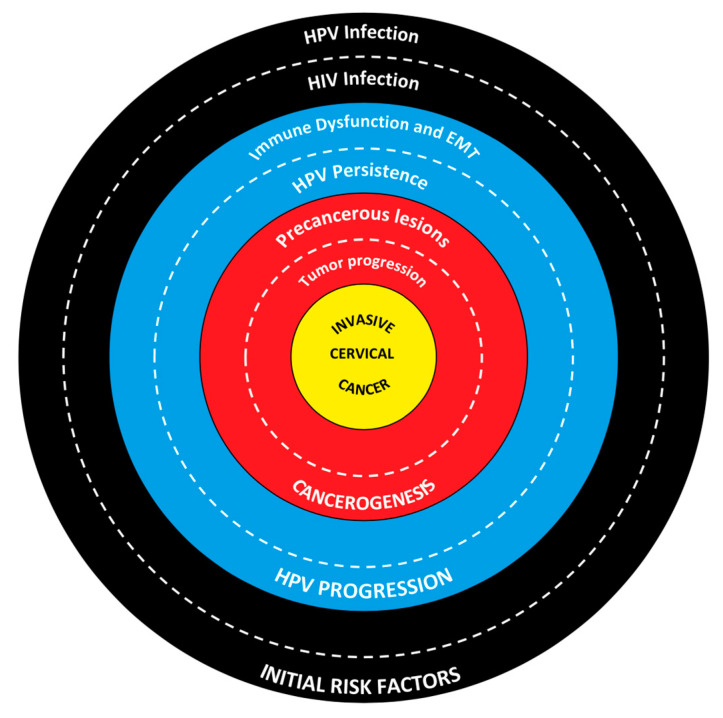
Interplay between HIV and HPV in cervical cancer development.

**Table 1 ijms-25-10358-t001:** HPV E proteins’ role in the viral cycle and oncogenesis in patients with HIV infection.

HPV Proteins	Role in Cervical Cancer	Association with HIV Infection	Impact on Disease Progression
E5	Enhances growth factor signaling and disrupts cellular processes, leading to proliferation and avoidance of apoptosis	Immunocompromised state due to HIV enhances the oncogenic potential of E5	Potentiates oncogenesis by leveraging compromised immune surveillance, contributing to more aggressive cancer progression
E6	Targets p53 for degradation, inhibiting apoptosis and allowing unchecked cellular replication	HIV-related immune suppression permits more extensive and unchecked activity of E6	Increases the likelihood of malignant transformation and cancer persistence in the host
E7	Disrupts cell cycle regulation by binding to pRb, releasing E2F transcription factors that stimulate cell proliferation	Weakened immune response in HIV-infected individuals fails to control E7-induced dysregulation	Leads to higher rates of cell turnover and progression to high-grade lesions and malignancy

**Table 2 ijms-25-10358-t002:** Interactions between HIV and HPV impacting cervical cancer progression.

Aspect	Description	Impact on Disease Progression
Immune system dysfunction	HIV impairs the immune system primarily by depleting CD4+ T cells, leading to apoptosis and reduced functionality. This suppression extends to specific T lymphocytes against HPV oncoproteins E6 and E7, crucial for controlling tumor progression in HPV-infected cervical cells	Increased susceptibility to high-grade squamous intraepithelial lesions (HSILs) due to reduced immune surveillance and inability to control HPV infection, thus facilitating progression to CC
Dendritic cell (DC) dynamics	HIV infection results in an increased number of immature DCs in cervical tissues, characterized by low expression of maturation markers CD83 and CD86. These immature DCs are less effective in antigen presentation and initiating a robust T-cell-mediated response.	Persistent HPV infection due to inadequate activation of immune responses against HPV, leading to sustained viral infection and increased risk of cervical lesion progression
Vaginal microbiome and inflammation	HIV-related changes in the vaginal microbiome lead to dysbiosis, characterized by an imbalance in microbial species that promotes a pro-inflammatory state. This inflammation can exacerbate HPV persistence and lesion progression.	Chronic inflammation associated with microbial dysbiosis enhances HPV persistence and promotes the progression of precancerous lesions to CC due to a continuously inflamed environment.
Epithelial–mesenchymal transition (EMT)	HIV proteins gp120 and Tat induce EMT in cervical epithelial cells. This process involves the loss of epithelial characteristics and gain of mesenchymal traits, facilitating tumor cell invasion, migration, and resistance to apoptosis.	Accelerates the progression of cervical lesions, increases the invasiveness and metastatic potential of CC cells, and may lead to resistance to conventional treatments

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
