# Peer review of "Entangled Connections: HIV and HPV Interplay in Cervical Cancer—A Comprehensive Review"

_ijms, 2024, doi:10.3390/ijms251910358_

Round 1

Reviewer 1 Report

Comments and Suggestions for Authors

This manuscript has summarized the functional interplay between HPV and HIV in terms of the malignancy of cervical cancer (CC) induced by HPV infection in women living with HIV (WLWH). The authors have cited the up-to-date findings and data trying to depict the complicated network of how HPV infection triggers the tumorigenesis in scenario of immunodeficiency, as well as outlining the current prevention, diagnosis and treatment of HPV-HIV coinfection and CC. This is a well-organized review which comprises sufficient background knowledge and updated discoveries. I do not have major concerns for the manuscript, but some minor points are listed for the authors’ consideration.

1 The section 4, “Interplay between HIV and HPV in cervical cancer development”, is overly long and dense containing too much information, which makes it difficult to read and digest. Since this section is the key of the manuscript, the authors may think to split it into sub-sections based on the contents.

2 The molecular interactions between HPV and HIV in page 6, lines 262-265, are weak compared to other parts, which needs to be enriched.

3 The sentences in page 7, lines 295-297 are a duplicate of lines 292-295.

Author Response

Comment 1. The section 4, “Interplay between HIV and HPV in cervical cancer development”, is overly long and dense containing too much information, which makes it difficult to read and digest. Since this section is the key of the manuscript, the authors may think to split it into sub-sections based on the contents.

Reply: We have reorganized Section 4 into two sub-sections to elucidate the two most relevant molecular mechanisms in carcinogenesis induced by the interaction between HIV and HPV: immune response dysfunction and epithelial-mesenchymal transition (EMT). Based on this, we have also revised the figure.

Comment 2. The molecular interactions between HPV and HIV in page 6, lines 262-265, are weak compared to other parts, which needs to be enriched.

Reply: We added few lines containing new information and references about you kindly pointed out

Comment 3. The sentences in page 7, lines 295-297 are a duplicate of lines 292-295.

Reply: We have removed the duplicate sentences.

Reviewer 2 Report

Comments and Suggestions for Authors

The study by Pavone et al. reviews the connection between HIV and HPV in cervical cancer development. Although the paper provides insights into the pathogenesis, prevention, and treatment of cervical cancer in women living with HIV, it needs minor revisions before publication.

Major comments

1. Section 3 "molecular mechanisms of oncogenesis in HPV-related cervical cancer" is not needed in this review (Lines 90-177). These topics have been covered by others. The section should focus on the viral proteins with roles linked to HIV infections and disease e.g., E5, E6 and E7. This section and table should be therefore shortened. The section also provides several scientific statements without proper citations.

2. A table summarizing the associations described in section 4 is needed and would be helpful for readers.

3. The review is missing components in the microenvironment that have an effect on both STIs, e.g., the vaginal microbiome. A brief discussion of the role of the microenvironment in HIV-HPV interactions (even if hypothesized) would be beneficial for this review. 

Minor comments

1. The abbreviation of cervical cancer was introduced in the text, but it is not used consistently (e.g., Lines 73, 77, 81, 85). Please update. 

2. The meaning for the abbreviation of DC-SIGN is missing (Line 206).

3. Proper references are missing between lines 199-220, and between lines 280-312.

4. Lines 225-234 describe findings on HPV mechanisms, but do not provide context related to HIV.

5. The abbreviation "oncHPV" is not used in the HPV field. Please update, and use the proper terminology: hrHPV. Lines 239, 245.

6. The mechanism of EMT is introduced in line 247, and continued later in line 269. Please organize this section, so it is easy for readers to understand the role of EMT in HPV/HIV. 

7. The oncoprotein E5 has a well-known role on EMT. A brief summary of these findings and how they relate to HIV is needed. 

7. Please update the term synergistic interaction in Figure 1. The immune system and HPV do not cooperate as far as we know. Both HPV and HIV alter the immune system. The figure description implies a timeline in which HIV infects the host first, weakens immune responses, thereby facilitating HPV persistence. The interaction would be between HIV and HPV. Please rephrase. 

8. Please use HPV DNA testing where appropriate, e.g., Line 337. 

Comments on the Quality of English Language

N/A

Author Response

Major comments

  1. Section 3 "molecular mechanisms of oncogenesis in HPV-related cervical cancer" is not needed in this review (Lines 90-177). These topics have been covered by others. The section should focus on the viral proteins with roles linked to HIV infections and disease e.g., E5, E6 and E7. This section and table should be therefore shortened. The section also provides several scientific statements without proper citations. Reply: We revised the paragraph, shortening it and adding only the data you requested. Also, we changed the table as you suggested.

  1. A table summarizing the associations described in section 4 is needed and would be helpful for readers. Reply: We added a table trying to summarize section 4, as you suggested.

  1. The review is missing components in the microenvironment that have an effect on both STIs, e.g., the vaginal microbiome. A brief discussion of the role of the microenvironment in HIV-HPV interactions (even if hypothesized) would be beneficial for this review. Reply: Thank to the Reviewer 2 for the insightful suggestion. We added a brief paragraph discussing the role of the vaginal microbiome and inflammation in modulating the immune response to HIV and HPV infections has been added to Section 4. This addition aims to enrich the discussion on the co-infection dynamics and immunoregulatory mechanisms involved.

Minor comments

  1. The abbreviation of cervical cancer was introduced in the text, but it is not used consistently (e.g., Lines 73, 77, 81, 85). Please update. Reply: We have replaced 'cervical cancer' with its abbreviation where necessary.

  1. The meaning for the abbreviation of DC-SIGN is missing (Line 206). Reply: We have spelled out the abbreviation of DC-SIGN on line 206.

  1. Proper references are missing between lines 199-220, and between lines 280-312. Reply: We have added the missing references

  1. Lines 225-234 describe findings on HPV mechanisms, but do not provide context related to HIV. Reply: We added the information you suggested.

  1. The abbreviation "oncHPV" is not used in the HPV field. Please update, and use the proper terminology: hrHPV. Lines 239, 245. Reply: Done.

  1. The mechanism of EMT is introduced in line 247, and continued later in line 269. Please organize this section, so it is easy for readers to understand the role of EMT in HPV/HIV.
  2. The oncoprotein E5 has a well-known role on EMT. A brief summary of these findings and how they relate to HIV is needed.

Reply to comment 6 and 7: We have reorganized the paragraph, focusing on the role of HIV and HPV in inducing EMT, ensuring clarity in how both viruses contribute to this process. Additionally, as suggested, we have provided a brief overview of the specific role of the HPV E5 oncoprotein in promoting EMT, emphasizing its molecular interplay with HIV within this mechanism.

7.bis Please update the term synergistic interaction in Figure 1. The immune system and HPV do not cooperate as far as we know. Both HPV and HIV alter the immune system. The figure description implies a timeline in which HIV infects the host first, weakens immune responses, thereby facilitating HPV persistence. The interaction would be between HIV and HPV. Please rephrase.

Reply: The reviewer's interpretation is correct. We have therefore modified the Figure 1 to emphasize that immune system dysfunction causing HPV persistence leads to the progression of the infection. We have also adjusted the caption accordingly.

  1. Please use HPV DNA testing where appropriate, e.g., Line 337. Reply: Done.
